# ON EXACT BIT-LEVEL REVERSIBLE TRANSFORMERS WITHOUT CHANGING ARCHITECTURES

## ABSTRACT

Various reversible deep neural networks (DNN) models have been proposed to reduce memory consumption in the training process. However, almost all existing reversible DNNs either require special non-standard architectures or are constructed by modifying existing DNN architectures considerably to enable reversibility. In this work we present the BDIA-transformer, which is an exact bit-level reversible transformer[1] that uses an unchanged standard architecture for inference. The basic idea is to first treat each transformer block as the Euler integration approximation for solving an ordinary differential equation (ODE) and then incorporate the technique of bidirectional integration approximation (BDIA) (originally designed for diffusion inversion) into the neural architecture, together with activation quantization to make it exactly bit-level reversible. In the training process, we let a hyper-parameter $\gamma$ in BDIA-transformer randomly take one of the two values $\{0.5, -0.5\}$ per training sample per transformer block for averaging every two consecutive integration approximations. As a result, BDIA-transformer can be viewed as training an ensemble of ODE solvers parameterized by a set of binary random variables, which regularizes the model and results in improved validation accuracy. Lightweight side information per transformer block is required to be stored in the forward process to account for binary quantization loss to enable exact bit-level reversibility. In the inference procedure, the expectation $\mathbb{E}(\gamma) = 0$ is taken to make the resulting architectures of BDIA-transformer identical to transformers up to activation quantization. Our experiments in both image classification and language translation show that BDIA-transformers outperform their conventional counterparts significantly in terms of validation performance due to the regularization effect of the set of $\gamma$ random variables while also requiring considerably less training memory.

## 1 INTRODUCTION

An active research trend in deep learning is to scale up the size of both the deep neural networks (DNNs) and the training data, with the aim of obtaining universal machine-learning models that are capable of accomplishing various tasks. Examples are large language models (LLMs) such as GPT-4 [1] and Llama 2 [26], which can, for example, have informative and friendly conversations with humans, solve mathematical problems, and produce high-quality source codes for programming tasks. A major bottleneck for training those large DNNs is that they often require large on-chip memory and large inter-chip communication bandwidth across many chips to accommodate both the DNN model and the intermediate activations of the input data-batch in back-propagation [8], which is referred to as *memory wall* in the literature.

One promising technique to alleviate the issue of memory wall is to design and train reversible DNNs [5; 6; 2; 19]. By doing so, the intermediate activations in the forward pass do not have to be stored in the memory to allow for back-propagation. Instead, they can be recomputed on-the-fly in the backward pass by exploiting the reversibility of the DNN, thus saving memory consumption by a large margin for deep DNNs. The procedure essentially reduces memory consumption at the cost of

---

[1]In fact, by following the same principle, we can also design exact bit-level reversible ResNets without changing its architecture for inference. Here, we focus on the transformer because of its popularity in the field of generative artificial intelligence.

a reasonable amount of additional computation. The reduction in memory also has the potential to improve throughput by increasing the batch size.

Early reversible DNNs, such as NICE [5], the methods of [20], Real NVP [6], and Inverse Autoregressive Flow [16] are differentiable transformations that were aimed at facilitating generative modeling. For an overview of such early normalizing flows see [17]. Inspired by these works, a range of reversible residual models were proposed subsequently, including RevNet [9], Glow [15], i-RevNet [13], i-ResNet [2], layerwise inversion [11], Fourier-transformation based CNN inversion [7], and Mintnet [23]. Another line of research enforces reversibility in deep learning from the perspective of ordinary differential equations (ODEs), which includes FFJORD [10], leapfrog networks [3], neural ODEs [4], momentum residual networks [21], and neural ODE inversion [25].

Recently, the research on reversibility has moved to other types of neural networks. The authors in [19; 34] proposed reversible vision transformers (referred to as RevViT) due to the popularity of LLMs. [27] utilized a reversible diffusion sampling method to optimize the noisy representation for the task of diffusion based image editing. To our best knowledge, all the above existing reversible DNNs either require non-standard architectures or are constructed by modifying the original DNN architectures considerably to enable reversibility.

In this paper, we propose BDIA-transformer, a new type of reversible transformer that uses an unchanged, standard architecture for the inference procedure. It is based on the bidirectional integration approximation (BDIA), which was recently proposed in [33] to enable diffusion inversion for round-trip image editing. To be able to incorporate BDIA into transformers for online back-propagation, we follow the common practice of treating each transformer block as an Euler integration approximation for solving an ordinary differential equation (ODE).

We make two main contributions in this work. Firstly, we propose BDIA-transformers by introducing a random hyper-parameter $\gamma \in \{-0.5, 0.5\}$ per transformer block per training sample to regularize the DNN models for improvement of validation performance. Each $\gamma$ parameter intends to average every two consecutive integration approximations. The training procedure becomes training of an ensemble of ODE solvers parameterized by a set of binary random variables. In the inference procedure, the expectation $\mathbb{E}[\gamma] = 0$ is utilized, which reduces BDIA-transformers to conventional transformers. Therefore, our method for enforcing reversibility is more flexible than existing ones.

Secondly, we perform activation quantization to allow for exact bit-level reversibility of BDIA-transformers. Note that the special setup of the $\gamma$ values in the subset $\{-0.5, 0.5\}$ when performing activation quantization leads to a 1 bit information loss per activation value per transformer block. Therefore, lightweight side information per transformer block needs to be stored during training to recover this 1 bit information loss. Despite this, the overall memory use is significantly reduced.

Experimental results on ViT for image classification and transformer for language translation show that the BDIA technique significantly improves the validation performance over that of the corresponding baseline transformers and simultaneously reduces training memory significantly. The improved performance results from the model regularization imposed by a set of $\gamma$ random variables. Experiments on text prediction demonstrate that BDIA-GPT2 significantly alleviates the over-fitting issue of GPT2 when intentionally trained on a very small dataset. Our empirical study also indicates that RevViT from [19] produces either inferior or comparable validation performance to that of its original counterparts.

## 2 RELATED WORKS

In recent years, various quantization strategies [30; 29; 28] have been proposed in the training and/or inference processes of DNN models on low-precision devices. For instance, the work [29] successfully performed quantization on DNN weights, activations, gradients and errors in the training and inference processes and obtained promising results. The recent work [18] demonstrated that LLMs with a quantization of 1.58 bits per model parameter exhibit comparable performance to non-quantized models. In summary, it was found that the validation performance of DNN models that incorporate those quantization operations is comparable with that of conventional DNN models. In our work, we only need to apply activation quantization to enable exact bit-level reversibility in training BDIA-transformers.

We note that in general, model quantization and design of reversible DNN models are two complimentary strategies for reducing memory consumption in the training process, where the first strategy

operates on the model parameters and the second one operates on the activation values. With the development of new model quantization methods such as [18], advancing the research frontier of reversible DNN models has become of great interest.

## 3 PRELIMINARY

**Neural networks as ODEs**: [4] highlighted the interpretation that passing a hidden state across layers that add a correction to that state can be viewed as Euler integration. While that paper identified architectures including residual nets and normalizing flows as following this pattern, it is equally true of diffusion models and transformers. This interpretation emphasizes the importance of considering more accurate integration schemes [14]. The need for improved integration is particularly apparent in round-trip image editing in diffusion models, where significant integration error will result in unintended visible alterations to the image.

**Diffusion sampling via solving ODE**: Recently, the work [33] proposed the BDIA technique to enable diffusion inversion for effective round-trip image editing. From a high level point of view, BDIA can be viewed as a time-reversible ODE solver. Given an initial diffusion state $z_T$ at time step $T$, the diffusion-based sampling process for generating realisic images $z_\epsilon$ at time $t = \epsilon > 0$ can be realized by solving a probability ordinary differential equation (ODE)

$$dz = d(z, t)dt \tag{1}$$

over the time interval $t \in [T, \epsilon]$. The gradient vector $d(z, t)$ includes the output of a pre-trained DNN model with $(z_t, t)$ as its input. The common practice for solving the above ODE is to first discretize the continuous time interval $[T, \epsilon]$ properly into a set of timesteps $\{t_i | i = 0, \ldots, N\}$ with $t_0 = T$ and $t_N = \epsilon$, and then perform certain integration approximation per small time-interval sequentially to compute the final diffusion state $z_N = z_\epsilon$.

**BDIA**: Suppose we would like to estimate the next diffusion state $z_{i+1} = z_{t_{i+1}}$ by solving (1) based on the recent information $(z_i, t_i)$ and $(z_{i-1}, t_{i-1})$, where $z_j = z_{t_j}$ for $j = i - 1, i$. The basic idea of BDIA is to compute $z_{i+1}$ by performing both the forward integration approximation $\Delta(t_i \rightarrow t_{i+1}|z_i) \left( \approx \int_{t_i}^{t_{i+1}} d(z, t)dt \right)$ and the backward integration approximation $\Delta(t_i \rightarrow t_{i-1}|z_i)$ $\left( \approx -\int_{t_{i-1}}^{t_i} d(z, t)dt \right)$ conditioned on $z_i$. One popular method for implementing $\Delta(t_i \rightarrow t_{i+1}|z_i)$ and $\Delta(t_i \rightarrow t_{i-1}|z_i)$ in the literature of diffusion models is by employing the DDIM update expression (see [22; 32; 33] for details). With the above two integration approximations, $z_{i+1}$ can be expressed as

$$z_{i+1} = z_{i-1} \underbrace{-(1 - \gamma)(z_{i-1} - z_i) - \gamma\Delta(t_i \rightarrow t_{i-1}|z_i)}_{\approx \int_{t_{i-1}}^{t_i} d(z,t)dt} + \underbrace{\Delta(t_i \rightarrow t_{i+1}|z_i)}_{\approx \int_{t_i}^{t_{i+1}} d(z,t)dt} \tag{2}$$

$$= \gamma z_{i-1} + (1 - \gamma)z_i - \gamma\Delta(t_i \rightarrow t_{i-1}|z_i) + \Delta(t_i \rightarrow t_{i+1}|z_i), \tag{3}$$

where $\gamma = (0, 1]$ averages $\Delta(t_i \rightarrow t_{i-1}|z_i)$ and $(z_{i-1} - z_i)$ for the time-slot $[t_{i-1}, t_i]$. The minus sign in front of the two quantities are due to the reverse integration direction. The quantity $-(z_{i-1} - z_i)$ is the previously computed integration approximation for $\int_{t_{i-1}}^{t_i} d(z, t)dt$. We note that negative $\gamma$ values are not applicable to the diffusion models considered in [33].

**On reversibility of BDIA**: The update expression (3) is carefully designed in [33] to enable diffusion inversion for round-trip image editing. By reformulating (3), $z_{i-1}$ can be easily computed in terms of $(z_i, z_{i+1})$. We note that due to the nature of the floating-point datatype, there might be error accumulation in round-trip image reconstruction, where the corresponding diffusion states in the forward and reverse process are not identical. In practice, error accumulation of BDIA in diffusion inversion is not a big issue [33] due to the fact that the number of timesteps is generally set to be small (e.g., 50 timesteps in either forward or reverse process) to make the time complexity reasonable.

One main difference between diffusion inversion and reversible transformers is that no gradient needs to be back-propagated in diffusion inversion for updating the DNN model. As a result, even if there is error accumulation in diffusion inversion, it is less severe than in reversible transformers where error accumulation in online back-propagation would slow down the training convergence or even make the training fail especially for very deep models like LLMs. In next section, we will explain how to

design exact bit-level reversible transformers in the training process to avoid any error-accumulation while at the same time, maintaining the architectures of the transformer in the inference procedure.

# 4 EXACT BIT-LEVEL REVERSIBLE TRANSFORMERS VIA BDIA

In this section, we first briefly review the transformer update expressions from the ODE viewpoint. We then propose the BDIA-transformer that enables exact bit-level reversibility with activation quantization. Specially, we will demonstrate how each of the two $\gamma$ values $\{-0.5, -0.5\}$ averages consecutive integration approximations. The training of a BDIA-transformer can then be interpreted as employing different ODE solvers for different training samples in a random manner. In addition, we explain why additional lightweight side-information is required to be stored to account for the binary quantization loss in online back-propagation.

## 4.1 REVISITING TRANSFORMER UPDATE EXPRESSION

A typical transformer block consists of the attention function (denoted as $\boldsymbol{f}(\cdot)$) and the function of feed-forward network (FFN) (denoted as $\boldsymbol{g}(\cdot)$), of which the trainable parameters are generally different for different block indices. Accordingly, the output $\boldsymbol{x}_{k+1}$ of the $k$th transformer block can be mathematically represented in terms of the input $\boldsymbol{x}_k$ as

$$\boldsymbol{x}_{k+1} = \boldsymbol{x}_k + \underbrace{\boldsymbol{f}_k(\boldsymbol{x}_k) + \boldsymbol{g}_k(\boldsymbol{x}_k + \boldsymbol{f}_k(\boldsymbol{x}_k))}_{\boldsymbol{h}_k(\boldsymbol{x}_k)}, \tag{4}$$

where we use $\boldsymbol{h}_k(\boldsymbol{x}_k)$ to denote the overall residual quantity that includes both the attention and FFN functions. For simplicity, we omit the pre-normalisation operations in (4), since these in fact do not affect the design of BDIA-transformers later on.

It is well known from the literature [4] that the $k$th forward step in (4) can be viewed as the Euler integration approximation of an ODE at timestep $t_k$:

$$\boldsymbol{x}_{k+1} = \boldsymbol{x}_k + \boldsymbol{h}_k(\boldsymbol{x}_k) = \boldsymbol{x}_k + \overbrace{\tilde{\boldsymbol{d}}(\boldsymbol{x}_k, t_k)(t_{k+1} - t_k)}^{\Delta(t_k \to t_{k+1}|\boldsymbol{x}_k)} \tag{5}$$

$$\approx \boldsymbol{x}_k + \int_{t_k}^{t_{k+1}} \tilde{\boldsymbol{d}}(\boldsymbol{x}, t)dt,$$

where $\tilde{\boldsymbol{d}}(\boldsymbol{x}_k, t_k)$ denotes the gradient vector with $(\boldsymbol{x}_k, t_k)$ as the input. Both $\tilde{\boldsymbol{d}}(\boldsymbol{x}_k, t_k)$ and $(t_{k+1} - t_k)$ are implicitly learned via the composite function $\boldsymbol{h}_k(\boldsymbol{x}_k)$, which is alternatively denoted as $\Delta(t_k \to t_{k+1}|\boldsymbol{x}_k)$.

As explained later on, we will introduce BDIA into the update expression of (5). Note that (5) is a general update expression that not only includes the transformer but also ResNet [12] and its variants. In fact, the following analysis can be easily adapted to enable reversibility of ResNet variants.

## 4.2 BDIA-TRANSFORMER WITHOUT QUANTIZATION

In this subsection, we first derive the update expressions of BDIA-transformer without quantization as an extension of (5), and then study the impact of the $\gamma$ values in $\{0.5, -0.5\}$. When $k = 0$, $\boldsymbol{x}_1$ can be computed by following (5) as

$$\boldsymbol{x}_1 = \boldsymbol{x}_0 + \boldsymbol{h}_0(\boldsymbol{x}_0) = \boldsymbol{x}_0 + \Delta(t_0 \to t_1|\boldsymbol{x}_0). \tag{6}$$

The update expression for $\boldsymbol{x}_{k+1}$ in the training process, $K - 1 \geq k \geq 1$, can be obtained by utilizing (2)-(3). Based on (5), we let

$$\Delta(t_k \to t_{k-1}|\boldsymbol{x}_k) = -\boldsymbol{h}_k(\boldsymbol{x}_k) \tag{7}$$

$$\Delta(t_k \to t_{k+1}|\boldsymbol{x}_k) = \boldsymbol{h}_k(\boldsymbol{x}_k). \tag{8}$$

By combining (7)-(8) and (2)-(3) with $i = k$, the update expression $\boldsymbol{x}_{k+1}$, $K - 1 \geq k \geq 1$, can be represented as

$$\boldsymbol{x}_{k+1} = \boldsymbol{x}_{k-1} + (1 - \gamma)(\boldsymbol{x}_k - \boldsymbol{x}_{k-1}) + (1 + \gamma)\boldsymbol{h}_k(\boldsymbol{x}_k) \tag{9}$$

$$= \gamma\boldsymbol{x}_{k-1} + (1 - \gamma)\boldsymbol{x}_k + (1 + \gamma)\boldsymbol{h}_k(\boldsymbol{x}_k), \tag{10}$$

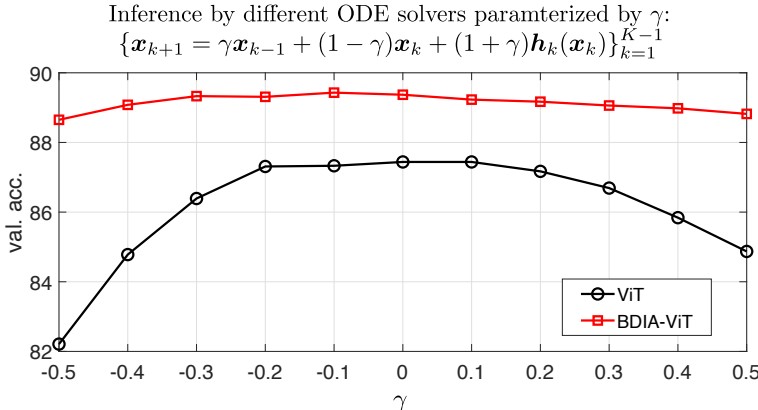

Figure 1: Validation performance of different ODE solvers parameterized by a single $\gamma$ parameter after training ViT and BDIA-ViT over CIFAR10. See Subsection 5.1 on how ViT and BDIA-ViT were trained. Each ODE solver in the inference procedure is realized by selecting $\gamma$ from $[-0.5, 0.5]$, which is fixed across all the transformer blocks for the same input image. The validation performance of BDIA-ViT is more robust than that of ViT.

where $\gamma$ is recommended to take values from $\{0.5, -0.5\}$ with equal probability per training sample per transform block, the impact of which will be explained in detail in the following. This is different from the work of BDIA-based diffusion inversion [33] where $\gamma$ has to be positive.

In the inference stage, $\mathbb{E}(\gamma) = 0$ is taken to replace $\gamma$ in (10), which leads to a simpler update expression that only involves $(\boldsymbol{x}_k, \boldsymbol{x}_{k+1})$:

$$\boldsymbol{x}_{k+1} = \boldsymbol{x}_k + \boldsymbol{h}_k(\boldsymbol{x}_k), \tag{11}$$

which is identical to the original transformer update expression (4)-(5).

**Impact of $\gamma$ parameter**: We now study the impact of the $\gamma$ parameter in (10). When $\gamma = 0.5$, it follows from (2)-(3) and (7)-(8) that the two integrations $\int_{t_{k-1}}^{t_k} \tilde{\boldsymbol{d}}(\boldsymbol{x}_\tau, \tau)d\tau$ and $\int_{t_k}^{t_{k+1}} \tilde{\boldsymbol{d}}(\boldsymbol{x}_\tau, \tau)d\tau$ of the transformer ODE are approximated as

$$\int_{t_{k-1}}^{t_k} \tilde{\boldsymbol{d}}(\boldsymbol{x}_\tau, \tau)d\tau \approx 0.5(\boldsymbol{x}_k - \boldsymbol{x}_{k-1}) + 0.5\boldsymbol{h}_k(\boldsymbol{x}_k) \tag{12}$$

$$\int_{t_k}^{t_{k+1}} \tilde{\boldsymbol{d}}(\boldsymbol{x}_\tau, \tau)d\tau \approx \boldsymbol{h}_k(\boldsymbol{x}_k), \tag{13}$$

where the integration over $[t_{k-1}, t_k]$ is computed as the weighted average of two consecutive integration approximations: $(\boldsymbol{x}_k - \boldsymbol{x}_{k-1})$ and $\boldsymbol{h}_k(\boldsymbol{x}_k)$.

When $\gamma = -0.5$, the two integrations $\int_{t_{k-1}}^{t_k} \tilde{\boldsymbol{d}}(\boldsymbol{x}_\tau, \tau)d\tau$ and $\int_{t_k}^{t_{k+1}} \tilde{\boldsymbol{d}}(\boldsymbol{x}_\tau, \tau)d\tau$ are approximated differently, given by

$$\int_{t_{k-1}}^{t_k} \tilde{\boldsymbol{d}}(\boldsymbol{x}_\tau, \tau)d\tau \approx (\boldsymbol{x}_k - \boldsymbol{x}_{k-1}) \tag{14}$$

$$\int_{t_k}^{t_{k+1}} \tilde{\boldsymbol{d}}(\boldsymbol{x}_\tau, \tau)d\tau \approx 0.5\boldsymbol{h}_k(\boldsymbol{x}_k) + 0.5(\boldsymbol{x}_k - \boldsymbol{x}_{k-1}). \tag{15}$$

In this case, the integration over $[t_k, t_{k+1}]$ is computed by averaging $(\boldsymbol{x}_k - \boldsymbol{x}_{k-1})$ and $\boldsymbol{h}_k(\boldsymbol{x}_k)$.

**Training via an ensemble of ODE solvers**: We use $\gamma_k$ to denote the random variable for the $k$th transformer block taking values in $\{0.5, -0.5\}$ with equal probability. The above analysis of (12)-(15) implies that each training sample goes through a particular ODE solver determined by sampling a set of $K - 1$ random variables $\{\gamma_k\}_{k=1}^{K-1}$, which specifies the integration path from the bottom transformer block until the top one. Due to randomness, different training samples will go through different ODE solvers. There are in total $2^{K-1}$ different ODE solvers, where each one corresponds to a unique integration path across the $K$ transformer blocks. Intuitively speaking, if we assume that

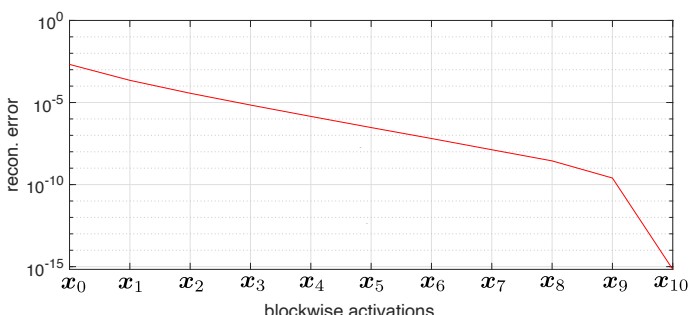

Figure 2: Demonstration of the accumulated reconstruction error by following (16) with the setup $\gamma_k \in \{0.5, -0.5\}$, $k = 1, \ldots, K-1$, when training BDIA-GPT2 with 12 transformer blocks.

each of the individual ODE solvers is well behaved (i.e., its training loss decays to a small value), then a convex combination will also converge to a small value.

In principle, different ODE solvers can also be applied in the inference procedure. For instance, one can set $\gamma$ to be constant within $[-0.5, 0.5]$ in (10) across all the transformer blocks for the same input. Fig. 1 demonstrates the validation performance of those different ODE solvers in the inference procedure for both trained BDIA-ViT and ViT over CIFAR10. It is clear that the validation performance of BDIA-ViT is much more insensitive to the single $\gamma$ parameter than that of ViT. The reason for the sensitivity of ViT to $\gamma$ in Fig. 1 is because only a single ODE solver is trained for ViT.

**On similarity to Dropout technique**: From a high level point of view, the training of BDIA-transformer is similar to the conventional dropout technique [24] to a certain extent. Dropout essentially attempts to train an ensemble of subnetworks of the entire DNN model when minimizing the objective function while BDIA-transformer intends to train an ensemble of different ODE solvers. In the inference procedure, an average of the ensemble of subnetworks in Dropout is utilized while in BDIA-transformer, the expectation $\mathbb{E}[\gamma_k] = 0$ is employed to replace $\gamma_k$.

**Remark 1.** *If reversibility is not of concern, one can freely specify the values for the random variables $\{\gamma_k\}_{k=1}^{K-1}$ in the training process for performance improvement as long as its distribution is symmetric around 0 such that $\mathbb{E}[\gamma_k] = 0$ for maintaining the original DNN architectures in the inference procedure. See Table 2 for ablation study of the impact of the $\{\gamma_k\}_{k=1}^{K-1}$ parameter on the validation performance when training BDIA-transformer for image classification.*

### 4.3 ON EXACT BIT-LEVEL REVERSIBILITY OF BDIA-TRANSFORMER WITH QUANTIZATION

As explained in Section 1, one strategy for reducing memory consumption in training transformer models like LLMs is to perform online back-propagation. That is, the intermediate activation outputs from the top transformer block until the bottom one are computed online when performing back-propagation to update the model. In this subsection, we first discuss the reversibility issue of the update expression (10. We then consider performing activation quantization to enable exact bit-level reversibility. We note that lightweight side information is required to be stored per transformer block for lossless online back-propagation.

**Limitation of the reversibility of (10)**: The update expression (10 is only theoretically reversible. That is, $\boldsymbol{x}_{k-1}$ can be computed in terms of $(\boldsymbol{x}_k, \boldsymbol{x}_{k+1})$ as

$$\boldsymbol{x}_{k-1} = \boldsymbol{x}_{k+1}/\gamma_k - (1-\gamma_k)/\gamma_k \boldsymbol{x}_k - (1+\gamma_k)/\gamma_k \boldsymbol{h}_k(\boldsymbol{x}_k), \qquad (16)$$

where we use $\gamma_k$ to indicate that different transformer blocks have their respective random variables. In practice, the setup $\gamma_k \in \{0.5, -0.5\}$, $k = 1, \ldots, K-1$, would lead to non-negligible error accumulation especially for very deep transformer models. The factor $\frac{1}{\gamma_k} = \pm 2$ in front of $\boldsymbol{x}_{k+1}$ would amplify the error when $k$ decreases from $K-1$ to 1, making the online back-propagation unstable. Fig. 2 illustrates that the reconstruction error indeed increases significantly when applying the online-back-propagation from the top transformer block until the bottom one.

**BDIA-transformer with quantization**: To allow for lossless online back-propagation, we propose to perform activation quantization. In particular, we use $\mathcal{Q}_l[\cdot]$ to denote the quantization operation to the bit-level precision of $2^{-l}$, given by

$$\mathcal{Q}_l[y] = \text{round}[y/2^{-l}]2^{-l}. \qquad (17)$$

Upon introducing $\mathcal{Q}_l[\cdot]$, the new update expression for BDIA-transformer can be represented as

$$x_0 \leftarrow \mathcal{Q}_l[x_0], \tag{18}$$

$$x_1 = x_0 + \mathcal{Q}_l[h_0(x_0)] \tag{19}$$

$$s_{k-1}[m] = \begin{cases} 1 & \text{if } \mathrm{mod}(x_{k-1}[m]/2^{-l}, 2) = 1 \\ 0 & \text{otherwise} \end{cases} \quad k \geq 1 \tag{20}$$

$$x_{k+1} = \mathcal{Q}_l[\gamma_k(x_{k-1} + s_{k-1}2^{-l})] + \mathcal{Q}_l[(1-\gamma_k)x_k + (1+\gamma_k)h_k(x_k)] \quad k \geq 1, \tag{21}$$

where $\gamma_k \in \{0.5, -0.5\}$, and $x_{k-1}[m]$ denotes the $m$th element of $x_{k-1}$. The $m$th element $s_{k-1}[m]$ indicates if the integer value $x_{k-1}[m]/2^{-l}$ is odd or not. It is immediate from (18)-(21) that $x_k = \mathcal{Q}_l[x_k]$ for all $K \geq k \geq 0$. That is, all the intermediate activation outputs $\{x_k\}_{k=0}^K$ have fix-point precision of $2^{-l}$.

Again in the inference procedure, we replace $\gamma_k$ in (21) by $\mathbb{E}(\gamma_k) = 0$. As a result, the update expression (21) can be simplified to be

$$x_{k+1} = \mathcal{Q}_l[x_k + h_k(x_k)] \quad k \geq 1. \tag{22}$$

The only difference of (22) w.r.t. the original transformer update expression (4) is that the quantization operation $\mathcal{Q}_l[\cdot]$ is performed for each activation output.

**On reversibility of (21) by storing lightweight side information**: We now consider recovering $x_{k-1}$ from $(x_k, x_{k+1})$ by utilizing (20)-(21). By using the fact that $\gamma_k \in \{0.5, -0.5\}$ and the definition for $s_{k-1}$, we can easily conclude that the quantity $\mathcal{Q}_l[\gamma_k(x_{k-1} + s_{k-1}2^{-l})]$ in (21) can be alternatively represented as

$$\mathcal{Q}_l[\gamma_k(x_{k-1} + s_{k-1}2^{-l})] = \gamma_k(x_{k-1} + s_{k-1}2^{-l}). \tag{23}$$

That is, the quantization operation has no effect on $\gamma_k(x_{k-1} + s_{k-1}2^{-l})$. This is because the vector $s_{k-1}$ essentially captures the 1-bit quantization loss of $\mathcal{Q}_l[\gamma_k x_{k-1}]$ per element.

Suppose in each forward pass in the training process, all the side information $\{s_{k-1}\}_{k=1}^{k=K-1}$ are stored in the memory. When we perform online back-propagation, each $x_{k-1}$ can be reconstructed losslessly in the form of

$$x_{k-1} = \frac{1}{\gamma_k}x_{k+1} - s_{k-1}2^{-l} - \frac{1}{\gamma_k}\mathcal{Q}_l[(1-\gamma_k)x_k + (1+\gamma_k)h_k(x_k)], \quad k \geq 1. \tag{24}$$

Consequently, the computed gradient in the online back-propagation will not be drifted from the ground truth, which is desirable in very deep transformer models.

**Remark 2.** *In principle, one can also make BDIA-transformer with $\{\gamma_k \in \pm 0.25\}_{k=1}^{K-1}$ to be exactly bit-level reversible. In this scenario, side information of 2 bits per activation value per transformer block is required to be stored to account for the quantization loss when performing online back-propagation. We omit the corresponding update expressions.*

**Limitations of BDIA-transformer**: To our best knowledge, all existing reversible DNN models do not need to store any lightweight side information in the forward process. The primary objective of most existing works is to design reversible DNN models (e.g., RevViT) that produce comparable validation performance as the original counterparts. From the perspective of memory reduction, existing reversible DNN models are more efficient than BDIA-transformer.

On the other hand, BDIA-transformer is designed to not only save training memory but also improve the generalization performance via model regularization. Our method is the first of its kind that attempts to maintain the transformer architecture in the inference procedure.

## 5 EXPERIMENTS

We evaluated BDIA-transformer for three different tasks: (1) training BDIA-ViT for image classification; (2) training BDIA-transformer for language translation; (3) training BDIA-GPT2 for text prediction. The hyper-parameter $l$ for quantization in all three tasks was set to $l = 9$. Three open-source repositories[2] were used in the experiments.

---

[2] https://github.com/kentaroy47/vision-transformers-cifar10
https://debuggercafe.com/language-translation-using-pytorch-transformer/
https://github.com/karpathy/nanoGPT

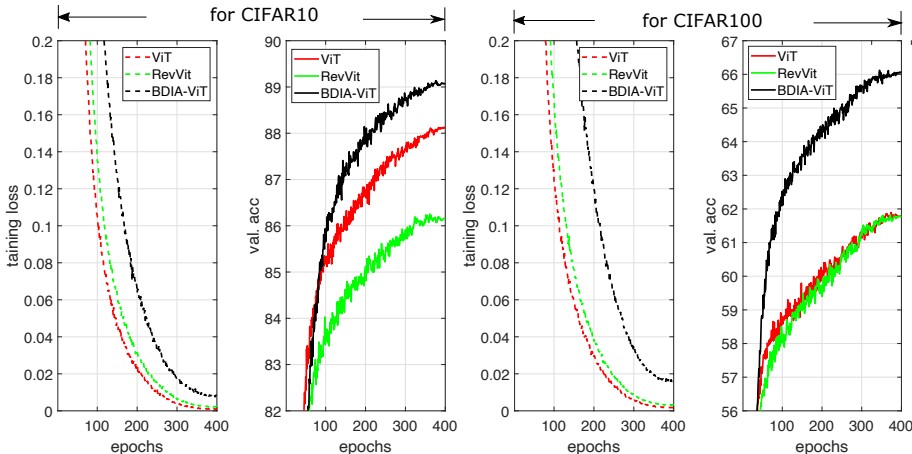

Figure 3: Performance comparison of ViT, RevViT [19], and BDIA-ViT for image classification over CIFAR10 and CIFAR100. $\{\gamma_k\}_{k=1}^{K-1}$ in the training procedure of BDIA-ViT were drawn from $\{\pm 0.5\}$ per training sample.

In brief, the obtained results from the first two tasks indicate that BDIA-ViT and BDIA-transformer significantly improves the validation performance of the original counterparts due to the model regularization effect of the random $\{\gamma_k\}_{k=1}^{K-1}$ variables. The results for the third task demonstrate that BDIA-GPT2 alleviates the overfitting issue of GPT2 significantly. RevViT from [19] was also tested for the first task. It was found that RevViT does not always improve the performance of ViT.

## 5.1 ON TRAINING BDIA-ViT

In this experiment, we trained BDIA-ViT with $K=6$ transformer blocks on CIFAR10 and CIFAR100. The performance of ViT and RevViT [19] was also evaluated to facilitate comparison. When implementing BDIA-ViT, the $\{\gamma_k\}_{k=1}^{K-1}$ parameters were drawn from $\{-0.5, 0.5\}$ with equal probability per training sample. In addition, we utilized the SET-Adam optimizer [31] in the training process with the configuration $(\eta_0, \beta_1, \beta_2, \epsilon) = (1e-4, 0.9, 0.999, 1e-18)$, where $\eta_0$ denotes the initial learning rate. The remaining training setups follow directly from the original open source (i.e., the first github link of footnote 2). Three experimental repetitions were performed for each training setup to mitigate the effect of the random seed.

Table 1: Validation accuracy and peak memory consumption for training three models over CIFAR10 and CIFAR100. $\{\gamma_k\}_{k=1}^{K-1}$ in the training procedure of BDIA-ViT were drawn from $\{\pm 0.5\}$ per training sample. The peak memory includes both the model parameters and the training states for a batchsize of 128.

| | RevViT [19] | | ViT | | BDIA-ViT | |
|---|---|---|---|---|---|---|
| | val. acc. | peak memory | val. acc. | peak memory | val. acc. | peak memory |
| CIFAR10 | 86.22±0.42 | 572.7MB | 88.15±0.55 | 1570.6MB | **89.10**±0.38 | 693.4MB |
| CIFAR100 | 61.89±0.31 | 572.7MB | 61.86±0.47 | 1570.6MB | **66.09**±0.80 | 693.4MB |

**Performance comparison**: Table 1 summarizes the obtained validation accuracy and the peak memory usages. It is clear that BDIA-ViT produces significantly higher validation accuracy than ViT and RevViT for both CIFAR10 and CIFAR100. Fig. 3 further visualizes the training and validation curves. It is seen that even though the training loss of BDIA-ViT is higher than the other two models across all the epochs, the validation accuracy improves remarkably in the end of training. This indicates that training an ensemble of ODE solvers parameterized (see Subsection 4.2) by $\{\gamma_k\}_{k=1}^{K-1}$ indeed regularizes BDIA-transformer properly.

In contrast, RevViT yields either inferior or comparable validation performance to ViT (see Table 1). This may be because RevViT modifies the architectures of ViT considerably to enable reversibility and therefore implicitly imposes uncontrolled regularization on the original transformer model. On the other hand, the regularisation introduced in BDIA-transformer is motivated by averaging consecutive integration approximations of an ODE in the original transformer, which leads to consistent performance gain across different datasets.

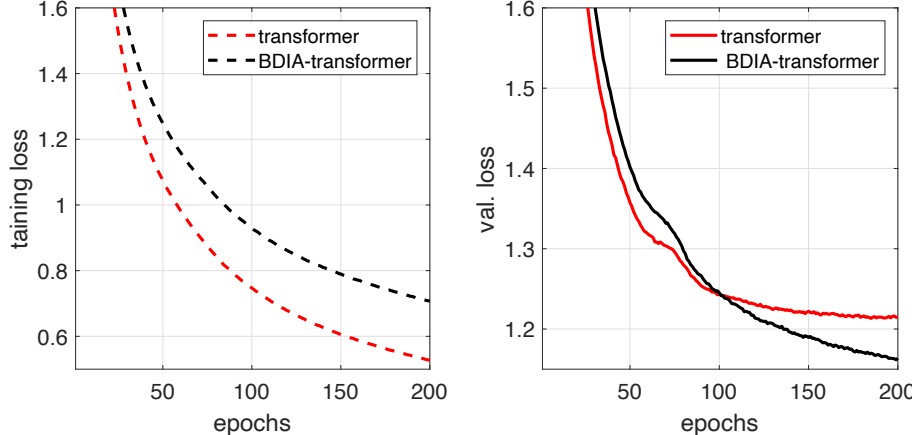

Figure 4: Performance comparison for English to French translation. $\{\gamma_k\}_{k=1}^{K-1}$ in the training procedure of BDIA-ViT were randomly drawn from $\{\pm 0.5\}$ per training sample.

It is also clear from Table 1 that RevViT is most memory-efficient. BDIA-ViT needs slightly more memory than RevViT because it needs to store the lightweight side information $\{s_k\}_{k=0}^{3}$ of the first 4 transformer blocks. BDIA-ViT improves the validation performance of RevViT at the cost of additional memory for storing the side information. We can also conclude from the table that online back-propagation does indeed significantly reduce the training memory.

**Ablation study regarding the $\{\gamma_k\}_{k=1}^{K-1}$ parameters**: As we mentioned in Remark 1, if reversibility is not of primary concern, different values for $\{\gamma_k\}_{k=1}^{K-1}$ can be employed for model regularisation. We conducted ablation study by evaluating the impact of $\gamma_k \in \{0.0, \pm 0.25, \pm 0.5, \pm 0.6\}$, $k = 1, \ldots, K-1$, on the validation performance of BDIA-transformer. For doing so, both the quantization and online back-propagation operations were turned off in BDIA-transformer. We emphasize that the non-zero $\gamma_k$ values were only utilized in the training process. At the inference stage, $\mathbb{E}[\gamma_k] = 0$ was used for computing the validation accuracy.

Table 2: Impact of the $\{\gamma_k\}_{k=1}^{K-1}$ parameter in BDIA-ViT (w.o. quantization and w.o. online back-propagation) on the validation accuracy in percentage.

| $\{\gamma_k\}_{k=1}^{K-1}$ | 0.0 | $\{\pm 0.25\}$ | $\{\pm 0.5\}$ | $\{\pm 0.6\}$ |
|---|---|---|---|---|
| CIFAR10 | 88.15±0.55 | 88.79=3±0.29 | **89.12**±0.22 | 88.89±0.15 |

Table 2 summarizes the obtained validation accuracy for different setups of the $\{\gamma_k\}_{k=1}^{K-1}$ parameters. It is clear that when $\{|\gamma_k| > 0\}_{k=1}^{K-1}$, the performance of BDIA-transformer improves considerably in comparison to that of the conventional ViT (i.e., corresponding to BDIA-ViT with $\{\gamma_k = 0.0\}_{k=1}^{K-1}$). The setup of $\{\gamma_k = \pm 0.5\}_{k=1}^{K-1}$ performs the best. In general, the larger the magnitude of the $\{\gamma_k\}_{k=1}^{K-1}$ parameters in $[0, 0.6]$, the slower the training speed (see Fig. 3). In practice, one can tune the magnitude of the $\{\gamma_k\}_{k=1}^{K-1}$ parameters within $[0, 0.6]$ to balance the trade-off between the training speed and the validation performance.

As we mentioned in Remark 2, side information can also be introduced for the setups of $\{\gamma_k = \pm 0.25\}_{k=1}^{K-1}$ to make BDIA-transformer reversible. However, side information of 2 bits per activation value per transformer block is required instead of 1 bit for $\{\gamma_k = \pm 0.5\}_{k=1}^{K-1}$. On the other hand, it is complicated to enforce exact bit-level reversibility for $\{\gamma_k = \pm 0.6\}_{k=1}^{K-1}$.

## 5.2 ON TRAINING BDIA-TRANSFORMER FOR ENGLISH-FRENCH TRANSLATION

Language translation is a classical natural language processing (NLP) task where the transformer shows a large performance gain over other DNN models. We adopted an existing open-source repository (i.e., the second github link of footnote 2) in our experiment. The tested BDIA-transformer has six transformer blocks in both the encoder and decoder, respectively. The BDIA update expressions

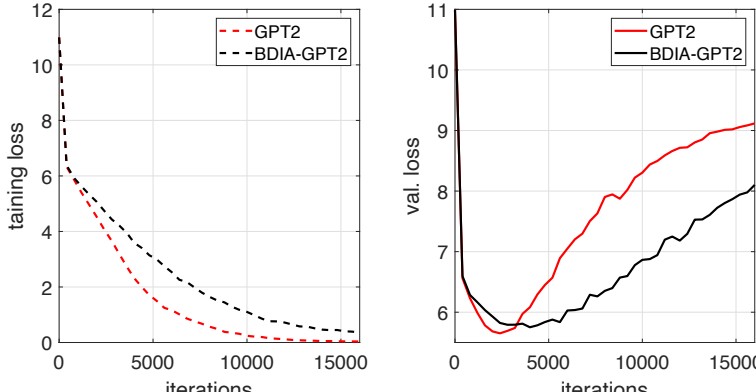

Figure 5: Performance comparison when training GPT2. $\{\gamma_k\}_{k=1}^{K-1}$ in the training procedure of BDIA-ViT were randomly drawn from $\{\pm 0.5\}$ per training sample.

were implemented in both the encoder and decoder. The performance of the conventional transformer was tested as a reference.

Fig. 4 visualizes the training and validation losses of the two tested models. It is clear that the obtained curves in Fig. 4 exhibit similar properties to those in Fig. 3. That is, even though the training loss of BDIA-transformer is higher than the conventional transformer across all the epochs, its validation loss is significantly lower after epoch 200.

### 5.3 ON TRAINING BDIA-GPT2

In this experiment, we consider training BDIA-GPT2 over the dataset of openwebtext by utilizing the third github link of footnote 2. Our primary objective for this task is to find out if BDIA can help to alleviate the over-fitting issue of GPT2 (we omit the phase "nano" for simplicity) for a very small training dataset. In doing so, we only took a small (i.e., 0.05%) subset from the entire dataset when training the model. The performance of GPT2 was evaluated as a reference. Both models have 12 transformer blocks.

Fig. 5 shows the training and validation curves for the two models. Similarly to Fig. 3-4, BDIA-GPT2 exhibits a slower training speed than GPT2. Considering the validation performance, the two validation curves for the two models exhibit the over-fitting issue. GPT2 produces the lower validation loss in the middle of the training. However, at the end of training, the validation loss of BDIA-GPT2 is significantly lower than that of GPT2. This demonstrates that BDIA indeed alleviates the over-fitting issue of GPT2 for a very small training dataset.

## 6 CONCLUSIONS

In this work, we have proposed the BDIA training technique to assist the training procedure of transformers. Firstly, each transformer block is taken as the Euler integration approximation for solving an ODE. The BDIA technique is then applied to average every two consecutive integration approximations in a transformer as a regularizer via a set of random variables $\{\gamma_k\}_{k=1}^{K-1}$, one variable per transformer block. Exact bit-level reversibility for lossless online back-propagation can be achieved for BDIA-transformer by performing activation quantization and storing only lightweight side information. The side information accounts for the binary quantization loss incurred by the special setup of $\{\gamma_k = \pm 0.5\}_{k=1}^{K-1}$. In the inference procedure, the expectation $\mathbb{E}[\gamma_k] = 0$ is taken to replace $\gamma_k$ in the update expression of BDIA-transformer, which reduces the update expression to that of the conventional transformer up to activation quantization.

Experiments on image classification and language translation show that BDIA-transformer produces significantly better validation performance than the corresponding baseline transformers while, at the same time, reducing training memory by performing online back-propagation. Experiments on text prediction indicate that BDIA-GPT2 alleviates the over-fitting issue of GPT2 over a very small training dataset significantly. In comparison, the experiments with RevVit demonstrate that it yields either inferior or comparable validation performance to that of ViT.

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
