# OpenReview forum: "On Exact Bit-level Reversible Transformers Without Changing Architectures"
_ICLR.cc/2025/Conference — ICLR 2025 Conference Withdrawn Submission_

### Official Review · Reviewer_r442 · 2024-10-23

**Soundness:** 4
**Presentation:** 3
**Contribution:** 3
**Rating:** 6
**Confidence:** 3

**Summary:**

The article derives a new method to make a transformer exactly reversible, based on a method initially dedicated to diffusion models and on the quantization of the activations. The proposed method also seeks to regularize the training of the model. Experiments are conducted on a variety of benchmarks with different transformer-based models, that is, vision-transformer, vanilla transformer and GPT-2. The proposed method achieves SOTA results on a vision task, while also preventing overfitting.

**Strengths:**

+ The paper is well-written, the ideas are clearly presented
+ Several insights on how the method works and relates to previous work are provided such as its similarity to dropout
+ The experiments effectively support the claims made in the article, such as the regularization effect of the method

**Weaknesses:**

+ The related work section is very short and surprisingly focuses on quantization, while quantization appears more in this work as a convenient trick to make information representation simpler rather than the primary topic of the article. Notably, the presentation of other works on reversible networks is missing (in this section)
+ The whole section on quantization (section 4.3) appears a bit messy. For instance, there is a parameter $l$ which is never really discussed, making it seem quite arbitrary. Similarly, the information storage vector $s_k$ is defined in a technical manner, and the section would benefit from providing more intuitive insights into its function.

**Questions:**

+ There is a tipo at line 169 (written $\\{-1/2, -1/2\\}$ instead of $\\{-1/2,1/2\\}$)

---

### Official Review · Reviewer_iNrX · 2024-11-07

**Soundness:** 1
**Presentation:** 2
**Contribution:** 2
**Rating:** 3
**Confidence:** 3

**Summary:**

The paper proposes a novel type of reversible transformers with the aim to reduce the memory during training. To this end, this work treats each transformer block as the Euler integration approximation in a manner similar to Neural ODEs. There are two main contributions. Firstly, the authors borrow a technique from recent works on diffusion inversion for round-trip image editing, which involves bidirectional integration approximation. This approximation introduces a hyperparameter $\gamma$. The authors propose selecting $\gamma$ randomly either -0.5 or 0.5 for each training sample and training block. Consequently, the training can be viewed as an ensemble of ODE solvers. This regularization led to observed improvements on validation data. Secondly, to ensure reversibility, the authors propose performing activation quantization while storing side information. This approach is validated on small datasets involving image classification, machine translation, and language modeling.

**Strengths:**

- The paper is generally well-written.
- The paper addresses an important and timely problem: reducing the memory consumption during the training of transformers, which is particularly relevant given the current widespread use of transformer models.
- The proposed idea is compelling as it retains the original architecture of transformers. This stands in contrast to existing approaches that typically involve modifications to the transformer architecture.

**Weaknesses:**

-	The reproducibility is low as there is no source codes or pseudo codes or detailed algorithms.
-	Although the paper includes thorough mathematical derivations, these seem to be more aligned with concepts from residual networks (ResNets) rather than focusing specifically on transformers. Notably, in equation (4), the authors treat the combined attention and feed-forward network modules as a residual term, resulting in derivations similar to those found in NeuralODEs with ResNets. However, these modules are key differentiators in transformer architectures compared to other models.
-	The experiments mainly consider small datasets or relies on toy examples for transformers.

**Questions:**

-	In figure 1, and in line 287, how did the authors integrate $\gamma$ into standard transformers?
-	In figure 2, the authors should the reconstruction errors w.r.t. the proposed method using quantization and side information. Otherwise, it is not clear the effectiveness of these tricks.
-	The authors should compare experimentally the proposed methods against vanilla transformers applied with dropout.
-	What dataset did the authors use in the machine translation experiments.
-	Although the authors show the memory gains, they should show the convergences in terms of wall-clock time to see better the computational complexity introduced by the proposed method.

---

### Official Review · Reviewer_Jk9h · 2024-11-08

**Soundness:** 2
**Presentation:** 4
**Contribution:** 2
**Rating:** 5
**Confidence:** 2

**Summary:**

The presented paper proposes a new training algorithm for transformers enabling reversibility up to a fixed precision level. While the paper itself focuses on transformers, the method seems to be applicable to any residual architecture. The method itself enables exact reversibility up to a given precision level, without architecture modification, and is thus broadly applicable. The authors introduces a regularizing parameter $\gamma$ which substantially alleviates overfitting issues in Transformers.

**Strengths:**

1. The paper is well-written and the algorithm is pretty easy to follow.
2. The experiments, while being rather small scale, do demonstrate a surprising regularization effect, able to tackle the overfitting issue of transformers.

**Weaknesses:**

1. The parameters $\gamma_k$ seems to already exist in the original BDIA paper and it is thus not clear what is the novelty of this paper on the matter.
2. The exact reversibility seems to require the quantization step but there is no ablation on the precision level $l$.
3. The small scale experiments does demonstrate a surprising regularization effect. However, the paper seems to seek reversibility, which is a feature usually used to scale up the model size given a fixed compute setup. Therefore, it is quite strange to focus on small scale experiments.

**Questions:**

Could the authors elaborate on the three weaknesses ?

---

### Official Review · Reviewer_g9w5 · 2024-11-08

**Soundness:** 2
**Presentation:** 1
**Contribution:** 3
**Rating:** 3
**Confidence:** 2

**Summary:**

The paper introduces BDIA-transformers, a novel approach to reversible transformers designed to reduce memory consumption during training without altering the architecture during inference. The approach leverages the Bidirectional Integration Approximation (BDIA) technique, which treats each transformer block as an Euler integration approximation for solving ordinary differential equations (ODEs). The experimental results show that BDIA-transformers outperform standard transformers in image classification and language translation tasks while reducing training memory requirements. For text prediction, BDIA-GPT2 prevents overfitting when trained on small datasets.

**Strengths:**

- The paper addresses the “memory wall” challenge, a critical issue in deep learning in which training large models requires extensive memory to store intermediate activations alongside the model itself. Authors explore reversible deep learning to tackle this issue. Reversible architectures allow backpropagation without storing or with minimal storage of intermediate activations, thus offering significant memory savings. The authors propose a novel technique by treating each transformer block as an Ordinary Differential Equation (ODE) solver and applying the Bidirectional Integration Approximation (BDIA) to these blocks.

- Experimental results show that BDIA-transformers improve performance across various tasks, including image classification and language translation. Authors claim that performance gains and memory efficiency are gained by implementing this method on Vision Transformers.

- This approach has the potential to enable the training of larger, more complex models when memory is the bottleneck, but we have very strong computational power.

**Weaknesses:**

- The paper is hard to follow, especially in the preliminary and method sections. It doesn’t clearly explain key ideas, like why transformers can be viewed as ODE solvers or how BDIA transformation is applied, which makes it confusing.

- The related work section mainly lists past studies without explaining how the field has developed or why this approach is needed. This makes it hard to see where this paper fits in with previous work.

- Some arguments supporting the method are vague. For instance, in the subsection "on similarity to dropout technique".

Overall, a clearer, simpler structure and explanations would make the paper easier to understand, especially for readers unfamiliar with reversible deep learning.

- The paper attributes BDIA-transformer's improved performance to the regularization effect of random variables, which is said to work similarly to dropout. However, it’s not clear if this is the only reason for the performance gains. The authors haven’t compared BDIA against other standard regularization methods on the baseline transformer, leaving open the chance that similar improvements could be achieved without the added complexity of BDIA.

- For language models, benchmarks and perplexity provide more informative insights into model quality. Could the authors test their methods against GPT-2 using additional metrics?

**Questions:**

- How does the extra computation time for a BDIA transformer compare to that of a regular transformer? In what situations does this additional overhead make sense? Could the authors also report the wall-clock performance of their method?

- Could the authors explain why they view a sequence of transformer blocks as a single ODE solver? This idea seems to fit better with diffusion models, as proposed by Zhang et al. 2023, but it’s not as clear for transformers.

- How well does the BDIA-transformer handle larger datasets or bigger models? Would the memory and speed trade-offs still hold up in these cases?

---

### Official Review · Reviewer_ovSj · 2024-11-09

**Soundness:** 3
**Presentation:** 2
**Contribution:** 3
**Rating:** 6
**Confidence:** 3

**Summary:**

This paper presents BDIA-transformer, a novel approach to creating reversible transformers that maintain their original architecture during inference. The authors incorporate the recently proposed bidirectional integration approximation (BDIA) technique, originally proposed for diffusion inversion, into the transformer architecture training process.

During the training process, a hyper-parameter $\gamma \in \{0.5, -0.5\}$ is randomly selected for each sample and transformer block to average consecutive integrations. This approach effectively trains an ensemble of ODE solvers. At inference time, the expectation $E(\gamma) = 0$ is used, reducing the model to a standard transformer with quantized activations.

The authors observed that the BDIA update expression is only theoretically reversible when using floating-point arithmetic, leading to error accumulation, especially for deep networks. To address this issue and enable lossless online backpropagation, they apply activation quantization to achieve exact bit-level reversibility.

Experimental results on various tasks, including image classification, language translation, and text prediction, demonstrate that BDIA-transformer (1) uses significantly lower overall memory during training (compared to ViT) and (2) acts as a regularizer, improving validation accuracy (over both RevViT and ViT) and reducing overfitting on small datasets.

**Strengths:**

- The background on the BDIA technique and its application in diffusion inversion is well-written and provides a natural motivation for its application in transformers.
- The authors provide a clear motivation for their work and the necessity of reversible transformer architectures.
- The application of this technique to transformers is novel and well-motivated.
    - More specifically, the connection between diffusion inversion and neural ODEs is an interesting one and provides a unique perspective on the problem.
- Experimental results demonstrate the effectiveness of the proposed BDIA-transformer on various tasks.

**Weaknesses:**

My main concerns with the paper are with the presentation and clarity of the methodology. Specifically:

- The paper begins (both in the abstract and introduction) with heavily references to the $\gamma$. However, the significance or meaning of this parameter is not immediately clear to the reader.
- The choices for specific values of $\gamma$ are not well-motivated. Further, for each $\gamma$ choice, seemingly different amounts of side-information are required, but these details are also not explained well.
- Similar to the first point, the reason for the existence of the activation quantization is not clear until much later in the paper. This is also something that should be explained much earlier in the paper (e.g., in the introduction).

**Questions:**

- Can you provide more intuition for the choice of $\gamma$ values? Why are these values chosen, and what do they represent? Further, do you have any theoretical insights into the choice of these values?
- Can you clarify, more explicitly, how you arrived at 1 bit for $\gamma \in \{0.5, -0.5\}$ and 2 bits for $\gamma \in \{0.25, -0.25\}$? How does this relate to the $\mathbf{s}_{k-1}[m]$ variable from equation (20)?
- In section 5.1, you mention hat you're training with $K=6$ transformer blocks. Later, when talking about the memory overhead, however, you denote the side information as $\{s_k\}_{k=0}^3$. Can you clarify this? Why is the side information only stored for the first 4 blocks?

---

### Official Review · Reviewer_6qsG · 2024-11-11

**Soundness:** 3
**Presentation:** 3
**Contribution:** 3
**Rating:** 5
**Confidence:** 2

**Summary:**

The paper proposes BDIA-transformer, a novel type of reversible transformer based on the bidirectional integration approximation (BDIA). A random hyperparameter $\gamma$ is introduced per transformer block per training sample to regularize the models. The paper further performs activation quantization to allow for exact bit-level reversibility of BDIA-transformers. Empirical results show that the BDIA technique outperforms baseline transformers and reduces training memory for image classification and language translation.

**Strengths:**

* Clarity: The authors clearly define the problem, related work, and their proposed method with well-written equations.

* Significance: The paper proposes a unique approach to achieve bit-level reversibility in transformers without architectural changes, leveraging techniques from ODE solvers and quantization. The introduction of the binary random variables also serves as a good regularization strategy.

**Weaknesses:**

* Lack of experiments: The paper compares BDIA-transformers with standard transformers and RevViT. It would benefit from a broader evaluation against other reversible architectures or quantization methods.

* The reliance on storing lightweight side information for exact reversibility might reduce its practical applicability as depth increases. It would be nice if this trade-off between memory efficiency and information storage is more analyzed.

**Questions:**

* What are the failure modes of BDIA-transformers? In which scenarios can the BDIA-transformers underperform compared to other transformers?

* Could you also report the comparison of BDIA-transformers and other architectures in terms of training time and computational cost?

* Minor: In Figure 3, the y-axis label should be “training loss”

---

### Note · Authors · 2024-11-22

**Comment:**

We thank all the reviewers for their appreciation of the novelty of our new approach for designing reversible transformers that preserves architectures in the inference stage. Also we are happy that the reviewers notice the significant performance improvement of BDIA-transformer over transformer for the tested tasks on small-scale image classification and language translation.

Due to limited time to do large scale experiments in the rebuttal period,  we decide to withdraw the paper and address the comments for a future submission.

**Withdrawal Confirmation:**

I have read and agree with the venue's withdrawal policy on behalf of myself and my co-authors.